

# Assessment of genotype by environment and yield performance of tropical maize hybrids using stability statistics and graphical biplots

Dedy Supriadi[1], Yusuf Mufti Bimantara[2], Yuniel Melvanolo Zendrato[3], Eko Widaryanto[4], Kuswanto Kuswanto[4] and Budi Waluyo[4]

[1] Graduate School, Brawijaya University, Malang, East Java, Indonesia
[2] Plant Breeding and Biotechnology, Graduate School, IPB University, Bogor, West Java, Indonesia
[3] Agrotechnology, Faculty of Agriculture and Business, Satya Wacana Christian University, Salatiga, Central Java, Indonesia
[4] Department of Agronomy, Faculty of Agriculture, Brawijaya University, Malang, East Java, Indonesia

## ABSTRACT

**Background:** Enhancing maize grain yield in tropical regions faces significant challenges due to variability in agroclimate, soil conditions, and agroecosystems. Understanding genotype (G) by environment (E) interaction (GEI) in plant breeding is crucial for selecting and developing high-yielding genotypes adapted to diverse environments.

**Methods:** Ten maize hybrids, including eight candidates and two commercial varieties, were evaluated across ten environments in Indonesia using a randomized complete block design with three replications. The GEI effect and yield stability were assessed using stability statistics, additive main effects and multiplicative interaction model (AMMI), and genotype + genotype × environment (GGE) biplot methods.

**Results and Discussion:** Analysis of variance revealed a significant GEI effect, indicating differences in hybrid responses for grain yield (GY), allowing for stability analysis. G01 showed the highest GY based on the best linear unbiased prediction (BLUP) across environments. Correlation analysis indicated strong associations between stability statistics ($YS_i$ and $S^{(6)}$) and GY, aiding in the selection of high-yielding hybrids. The integration of AMMI with the BLUP method, and weighted average of absolute scores (WAASB), enabled precise measurement of genotype stability. Overall, G01 (R0211), G04 (R0105), G05 (R0118), and G07 (R0641) emerged as high-yielding, stable hybrids based on stability statistics, AMMI, GGE biplot, and WAASB rankings. These hybrids offer promising candidates for maize genetic improvement programs in tropical regions.

# INTRODUCTION

Maize (*Zea mays* L.) is a significant global food crop, playing a vital role in the international agricultural system. It is primarily used for animal feed, human

Corresponding author
Budi Waluyo, budiwaluyo@ub.ac.id

consumption, and various food products (*Erenstein et al., 2022*). Maize is the third most consumed cereal for human food (*FAOSTAT, 2022*) and holds high economic value in many Asian countries, including Indonesia (*Agus et al., 2019*). As an alternative food source, maize is projected to contribute substantially to future cereal demand. Therefore, innovations to increase maize productivity through genetic approaches, agronomic practices, land management strategies, and other technologies are crucial to meet food needs (*Albahri et al., 2023*). Sustainably increasing maize productivity requires a deep understanding of the factors that affect grain yield (*Rusinamhodzi et al., 2020*; *Kipkulei et al., 2024*).

Maize is influenced by various factors, including environmental conditions, growing seasons, climatic elements, and agronomic management, all of which impact maize growth and yield (*Azrai et al., 2023*; *Petrović et al., 2023*; *Zendrato, Suwarno & Marwiyah, 2024*). Understanding maize hybrid performance under different environmental conditions is essential for efficient selection (*Yue et al., 2022*; *Ljubičić et al., 2023*). However, selection under varying climates and environmental conditions is complex due to genotype-environment interaction (GEI). A hybrid that performs well in one environment may not succeed in another. Awareness of genotype-environment (G × E) interactions provides valuable insights for selecting maize hybrids that consistently deliver high yields across diverse conditions, helping mitigate grain yield variability, a significant challenge for farmers and breeders (*Konate et al., 2023*; *Singamsetti et al., 2023*).

Multi-environment trials (METs) are vital in breeding programs for analyzing genotype-by-environment interaction and assessing genotypes' adaptability and stability (*Lee et al., 2023*; *Kondombo et al., 2024*). It is crucial to use statistical methods to evaluate the yield stability of maize hybrids in diverse environments. Breeders use various statistical models to select the best genotypes for commercial varieties, addressing GEI challenges comprehensively (*Gela et al., 2023*; *Karimizadeh et al., 2023*). These models are especially useful in specific agro-climatic regions, including tropical areas, where they allow for in-depth genotype evaluation under diverse conditions (*Azrai et al., 2022*; *Matongera et al., 2023*).

Statistical stability methods are categorized into parametric and nonparametric types. Parametric methods, encompassing univariate and multivariate stability analyses, depend on distribution assumptions, while nonparametric methods use response variable averages and rankings to estimate values (*Pour-Aboughadareh et al., 2019*; *Gela et al., 2023*). Multivariate analyses, such as the multiplicative interaction model (AMMI) (*Gauch, 2013*) and the genotype + genotype × environment (GGE) biplot (*Yan et al., 2000*), interpret GEI effects through principal component analysis (PCA) for graphical depiction. *Olivoto et al. (2019)* introduced the weighted average of absolute scores (WAASB), which combines AMMI's graphical tools with the BLUP technique to identify high-yielding, stable genotypes (*Singamsetti et al., 2021*; *Yue et al., 2022*). Each method has strengths and weaknesses in addressing GEI phenomena. Relying on a single stability method to identify stable, high-yielding genotypes is less reliable (*Wicaksana et al., 2022*). Integrating stability statistics with graphical biplot methods provides a more comprehensive understanding of GEI, enabling the identification of maize hybrids with both high GY and stability in METs.

This approach is appropriate for datasets with statistical assumptions, including interaction effects with simple estimates based on ranked performance data, and it allows for the visual identification of response patterns across mega-environments through a graphical biplot. This study aimed to elucidate the response of maize hybrids in various environments and determine the stable and adaptive hybrids using multiple stability statistics in METs.

## MATERIALS AND METHODS

### Genetic material and experimental sites

The genetic material used in the multilocation test consisted of eight tested hybrids: R0211 (G01), R0J020 (G02), R0654 (G03), R0105 (G04), R0118 (G05), R0498 (G06), R0641 (G07), and R0J016 (G08), along with two commercial hybrids as checks, RSA002 (G09) and NK7328 (G10). G01 to G08 were single maize crosses from selected inbred lines developed by PT Restu Agropro Jayamas (RAJA), serving as candidates for a new superior variety. RSA002 was released by PT RAJA in 2020, while NK7328 (or NK SUMO) is a variety from Syngenta Indonesia that has been widely cultivated in tropical regions, particularly Indonesia. METs were conducted from March to October 2023 at ten maize centers in Indonesia (Table 1), including Klaten (E01), Bantul (E02), Tuban (E03), Boyolali (E04), Nganjuk (E05), Blitar (E06), Kediri (E07), Jombang (E08), Malang (E09), and Takalar (E10). Different climate types were categorized based on Oldeman's classification (C2, C3, D1, D2, D3), indicating agro-climatic variability in each environment. Annual rainfall across the experimental sites ranged from 1,279 to 2,030 mm. The minimum temperature was 19.17 °C in E10, while the maximum was 32.81 °C in E07. Rainfall and climate data were gathered from meteorological stations near the experimental sites. The ten locations differed in agro-ecologies, with a lowland zone (all locations except E09) at altitudes between 55 and 217 m above sea level (masl) and a mid-altitude zone at 658 masl (E09). This range of topographies and elevations offers a broad representation of the region's environmental conditions. The soil types found at the experimental locations included grumosol, ultisol, alluvial, andosol, and inceptisol. Alluvial and grumosol were predominated, with alluvial soils, formed from river deposits, being rich in minerals and having a clay loam texture, while grumosol had a clay or clay loam texture. Other soils, such as ultisol (E04) and inceptisol (E09), had clay textures, and andosol (E07) was notable for its volcanic ash content and dark color, with a sandy loam texture. The soil pH ranged from slightly acidic to nearly neutral. The differences in agroclimate and agro-ecology underscore the importance of environmental factors in evaluating maize hybrid performance, as these factors influence plant growth and development. The specific details for each environment are presented in Table 1.

### Experimental design and field management

This study was conducted using a randomized complete block design with three replications at each experimental site. Field preparation began with full tillage using a tractor to break up clumps, smooth the soil surface, and create furrows for planting. The harvested plot size was four rows, each 5.0 m long (14 m$^2$). Border rows were used at the

**Table 1 Description of the 10 environments used for maize hybrids evaluation in Indonesia.**

| Env. code | Location | Soil type | Soil texture | Altitude masl | Climate type* | Average annual rainfall (mm) | Temperature (°C) | |
|---|---|---|---|---|---|---|---|---|
| | | | | | | | Minimum | Maximum |
| E01 | Wonosari, Klaten, Central Java | Grumusol | Loam | 217 | C2 | 1,279 | 22.58 | 31.67 |
| E02 | Banguntapan, Bantul, Special Region of Yogyakarta | Grumusol | Clay loam | 80 | C3 | 1,275 | 22.41 | 31.17 |
| E03 | Jenu, Tuban, East Java | Grumusol | Loam | 116 | D3 | 1,731 | 22.24 | 31.37 |
| E04 | Teras, Boyolali, Central Java | Ultisol | Clay | 106 | C2 | 1,832 | 19.97 | 29.42 |
| E05 | Ngronggot, Nganjuk, East Java | Alluvial | Clay loam | 55 | D2 | 1,828 | 20.20 | 29.65 |
| E06 | Wonodadi, Blitar, East Java | Alluvial | Clay loam | 97 | C3 | 1,742 | 19.62 | 30.14 |
| E07 | Plosoklaten, Kediri, East Java | Andosol | Sandy loam | 70 | C3 | 1,330 | 22.16 | 32.81 |
| E08 | Bandar Kedungmulyo, Jombang, East Java | Alluvial | Clay loam | 148 | C3 | 1,390 | 23.48 | 31.23 |
| E09 | Tumpang, Malang, East Java | Inceptisol | Clay | 658 | C3 | 1,697 | 21.70 | 31.56 |
| E10 | South Galesong, Takalar, South Sulawesi | Alluvial | Clay loam | 56 | D1 | 2,030 | 19.17 | 29.51 |

**Note:**
* Oldeman's classification of climate types.

edges of each field to minimize edge effects. The planting distance was 70 cm between rows and 20 cm within rows. Two seeds were sown per hole, and seedlings were thinned to one plant per hole 10 days after planting (DAP). Fertilization was applied twice, at 10–14 DAP and 30–35 DAP, using NPK at 400 kg ha$^{-1}$ (15% N, 15% P, and 15% K) and urea at 350 kg ha$^{-1}$ (46% N). Fertilizer was applied by making a furrow approximately 10 cm from the row to ensure it was easily absorbed by the roots, minimizing leaching and evaporation. All crop management practices were followed according to the Indonesian Ministry of Agriculture's technical guidelines at each site. Weeding involved clearing weeds around the plants, and mulching was done by raising the mounds and loosening the soil for better aeration. Mulching also helped prevent root and stem bending in the maize hybrids. Irrigation was carried out during vegetative growth and flowering, up to early seed formation, depending on soil conditions when rainfall was insufficient. Other practices, such as ridging, pest control, and disease management, followed the technical protocol. Harvesting was done at physiological maturity, indicated by the black layer at the grain's base. GY at 15% moisture content, the study's main trait, was calculated using the following formula:

$$Yield\ (t\ ha^{-1}) = \frac{10000}{PS} \times \frac{100 - MC}{100 - 15} \times \frac{EW}{1000} \times SP$$

where PS is the harvested plot size (m$^2$), MC is the actual moisture content at harvest, EW is the ear yield per plot (kg), and SP is the shelling percentage.

## Statistical analysis

A joint analysis of variance (ANOVA) was conducted to assess genotype-by-environment interactions on GY, considering three factors: genotype (hybrids), environment, and

**Table 2 Stability parameters calculated in this study.**

| No | Statistic | Symbol | References |
|----|-----------|--------|------------|
| **Parametric** | | | |
| 1 | Coefficient of variation | CV | *Francis & Kannenberg (1987)* |
| 2 | Regression coefficient | $b_i$ | *Finlay & Wilkinson (1963)* |
| 3 | Deviation from regression | $S^2_{di}$ | *Eberhart & Russell (1966)* |
| 4 | Shukla'stability variance | $\sigma^2_i$ | *Shukla (1972)* |
| 5 | Wricke's ecovalence | $W^2_i$ | *Wricke (1962)* |
| 6 | Average of the squared eigenvector values | EV | *Sneller, Kilgore-Norquest & Dombek (1997)* |
| 7 | AMMI stability index | ASI | *Jambhulkar et al. (2017)* |
| 8 | AMMI stability value | ASV | *Purchase, Hatting & Van Deventer (2000)* |
| 9 | Sum of IPCs Scores | SIPC | *Purchase, Hatting & Van Deventer (2000)* |
| 10 | Modified AMMI stability value | MASV | *Adugna & Labuschagne (2002)* |
| 11 | The absolute value of the relative contribution of IPCs to the interaction | Za | *Zali et al. (2012)* |
| 12 | Weighted average of absolute scores | WAASB | *Olivoto et al. (2019)* |
| **Nonparametric** | | | |
| 1 | Kang's rank sum | $YS_i$ | *Kang (1988)* |
| 2 | Huehn's and Nassar and Huehn's statistics | $S^{(1,2,3,6)}$ | *Huehn (1990)*, *Nassar & Huhn (1987)* |
| 3 | Thennarasu's statistics | $NP^{(1-4)}$ | *Thennarasu (1995)* |

replication, using a linear model with interaction effects. Genotype and genotype-by-environment were treated as random effects. Heritability, variance components, and the coefficient of variation were also calculated.

Stability analysis was applied when genotype-environment interactions significantly affected GY. Fifteen stability statistics (Table 2), comprising both parametric and nonparametric parameters, were used to evaluate the stability and ranking of each hybrid across environments. Spearman's rank correlation was used to analyze associations among stability parameters. Additionally, multivariate analyses, including AMMI (AMMI I and AMMI II) and GGE biplot (discriminativeness *vs.* representativeness, mean *vs.* stability, genotype ranking, and which-won-where), were employed to assess hybrid stability and adaptation to the test environments. AMMI was used to decompose GEIs into principal components, with the analysis performed according to the following equation:

$$y_{ij} = \mu + \alpha_i + \tau_j + \sum_{k=1}^{p} \lambda_k a_{ik} t_{jk} + \rho_{ij} + \varepsilon_{ij}$$

where, $y_{ij}$ is GY response of the *i*-th genotype in the *j*-th environment; $\mu$ is grand mean; $\alpha_i$ and $\tau_j$ represent genotype effect *i*-th and the environment effect *j*-th; $\sum_{k=1}^{p} \lambda_k a_{ik} t_{jk} + \rho_{ij} + \varepsilon_{ij}$ is model of multiplicative genotype × environment interaction effect, which $\lambda_k$ is singular value for the *k*-th interaction principal component axis (IPCA), $a_{ik}$ is *i*-th genotype eigenvector for axis *k*, $t_{jk}$ is the *j*-th environment eigenvector for axis *k*; $\rho_{ij}$ stands for the residual not explained by the IPCAs used in the model; and $\varepsilon_{ij}$ is error relevant to the model (*Olivoto et al., 2019*).

**Table 3 Joint analysis of variance for grain yield.**

| Source | Df | Sum Sq | Mean Sq | F value | Pr (>F) |
|---|---|---|---|---|---|
| Environment (E) | 9 | 161.10 | 17.91 | 36.04 | 0.000 |
| Replication (R)/E | 20 | 43.40 | 2.17 | 4.37 | 0.000 |
| Hybrids (G) | 9 | 143.30 | 15.92 | 32.04 | 0.000 |
| G × E | 81 | 145.10 | 1.79 | 3.61 | 0.000 |
| Residuals | 180 | 89.40 | 0.50 | | |
| CV (%) | 5.81 | | | | |

**Note:**
CV, coefficient of variation.

All statistical analyses were carried out in R software 4.4.1 using the "metan" package (*Olivoto & Lúcio, 2020*) for joint ANOVA, heritability, AMMI, GGE biplot, and Spearman's rank correlation of stability parameters. Stability statistics were estimated using the web-based PBSTAT-GE 3.5 (www.pbstat.com) and the "metan" package. Violin plot visualizations were created using the "ggpubr" package, while BLUP plotting for hybrids was done using the "ggplot" package.

# RESULTS

## Joint analysis of variance and estimated variance components

The ANOVA for GY across individual environments showed significant genotype effects ($p < 0.05$), except for E09 (Tumpang-Malang) (Table S1). The combined ANOVA (Table 3) indicated a significant environment (E) effect, highlighting the diversity of the tested environments, which included variations in soil type, altitude, climate, and other factors influencing GY. Genotype (G) effects were also significant, suggesting that the maize hybrids had different yield responses. The GEI effect on GY was significant ($p < 0.001$), as seen in Table 4, based on the mixed-model likelihood ratio test. These results show varying responses of each maize hybrid to different environments. Figure 1 further illustrates the variation in the GY values of 10 maize hybrids under 10 environments, thereby contributing to genotype-by-environment interactions. The large interaction between genotype and environment enables the stability analysis of GY to determine the potential of stable hybrids across environments or adaptive hybrids in certain environments.

The estimated heritability for GY was divided into plot- and mean-based values (Table 4). The plot-based heritability ($h^2$) was 0.34% or 34%, while the mean-based heritability ($h^2_{gm}$) was 0.89% or 89%. The mean-based heritability was estimated by calculating phenotypic variance ($\sigma^2_g$) dividing the environmental variance ($\sigma^2_g$) and GEI variances ($\sigma^2_g$) by the number of environments and replications; this heritability was based on the mean of the hybrids. Genotype selection accuracy, considering the correlation between observed and predicted mean GY, was high ($As = 0.94$). However, the correlation of genotypic values across environments ($r_{ge}$) was only 0.47, attributed to high $CV_r$ (%) and residual variance ($\sigma^2_g$) relative to genotypic variance.

**Table 4 Estimated of the variance component of grain yield.**

| Statistic | Likelihood ratio test | |
| --- | --- | --- |
| | **G** | **GEI** |
| $x^2$ | 32.70 | 40.80 |
| $p$-value | 1.10E−08 | 1.03E−12 |
| REML | Estimation of variance components | |
| $\sigma^2_g$ | 0.47 (33.65%) | |
| $\sigma^2_{ge}$ | 0.43 (30.84%) | |
| $\sigma^2_e$ | 0.50 (35.51%) | |
| $\sigma^2_p$plot basis | 1.40 | |
| $\sigma^2_p$mean basis | 0.53 | |
| $h^2$plot basis | 0.34 | |
| $R^2_{gei}$ | 0.31 | |
| $h^2_{gm}$mean basis | 0.89 | |
| $A_s$ | 0.94 | |
| $r_{ge}$ | 0.47 | |
| $CV_g$ (%) | 5.66 | |
| $CV_r$ (%) | 5.81 | |
| $CV_g/CV_r$ ratio | 0.97 | |
| SE | 0.08 | |
| SD | 1.39 | |

**Note:**

Abbreviations: G, genotype; GEI, genotype × environment interaction; REML, restricted maximum likelihood; $\sigma^2_g$, genotypic variance; $\sigma^2_{ge}$, genotype-by-environment interaction variance; $\sigma^2_e$, residual variance; $\sigma^2_p$, phenotypic variance; $h^2$, broad-sense heritability; $R^2_{gei}$, coefficient of determination of the interaction effects; $h^2_{gm}$, heritability of the genotypic mean; $A_s$, accuracy of selection; $r_{ge}$, genotype-environment correlation; $CV_g$ (%), genotypic coefficient of variation; $CV_r$ (%), residual coefficient of variation; CV ratio, ratio between genotypic and residual coefficient of variation; SE, standard error; SD, standard deviation.

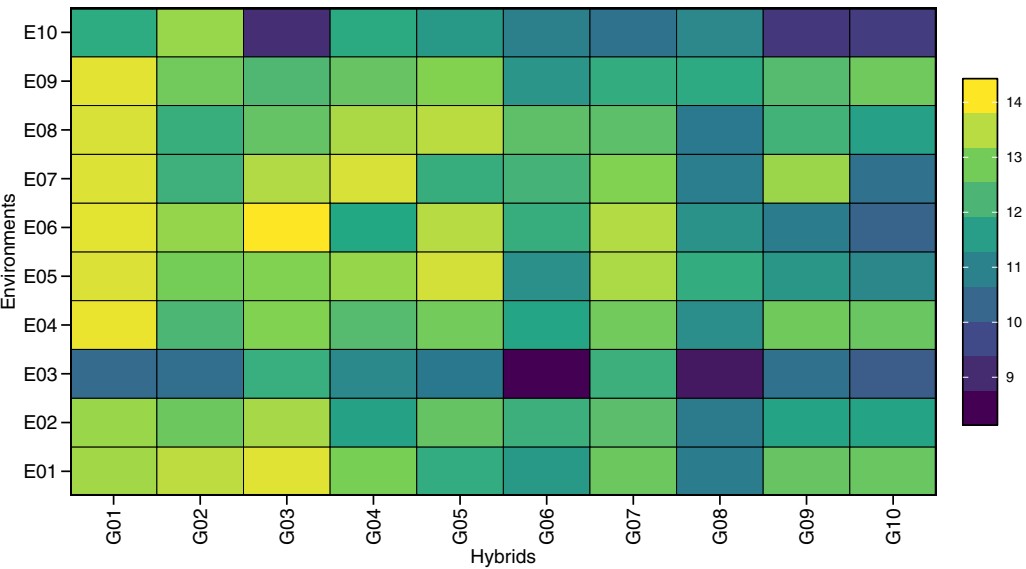

**Figure 1 The difference response plot of grain yield contributing to genotype *vs.* environment of 10 maize hybrids in 10 locations.**
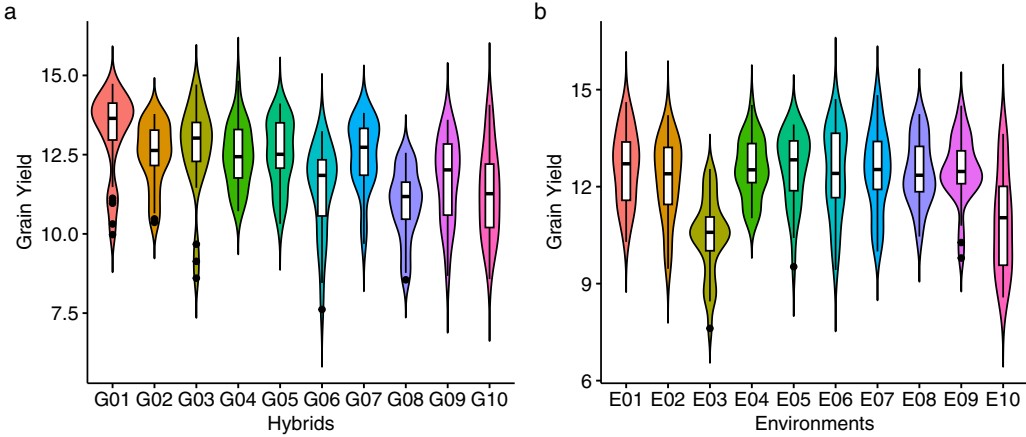

**Figure 2  Performance of grain yield.** The violin plot of genotypes (A) and environments (B) showing the distribution of grain yield performance.     

## Grain yield performance and predicted mean

Figure 2 presents the distribution of GY (t ha$^{-1}$) for 10 maize hybrids (Fig. 2A) and 10 environments (Fig. 2B). The violin plot highlights the degree of variation in GY based on genotype and location. Mean GY values for each hybrid in each environment are shown in Table S2. G01 had the highest GY (13.19 t ha$^{-1}$) across all environments, while G08 had the lowest (10.95 t ha$^{-1}$). Figure 2B shows that the lowest performance of all hybrids occurred in environment E03 (10.47 t ha$^{-1}$), while E01 had the highest mean GY, followed by E04 and E05. The coefficient of variation (CV) ranged from 4.14 to 9.16, with E10 showing the highest CV, indicating strong hybrid response variation in this environment. Figure 3 shows predicted mean GY for each hybrid using the BLUP model, confirming G01 (R0211) as the highest (13.07 t ha$^{-1}$) and G08 (R0J016) as the lowest (11.08 t ha$^{-1}$) (Table S3). The observed and predicted mean GY values were similar, as reflected by high genotypic selection accuracy ($As$ = 0.942, Table 4). Six hybrids (G01, G03, G05, G02, G07, and G04) had above-average yields, while four were below average.

## Yield stability estimation using stability statistics

GY was the primary quantitative trait used to assess the stability of maize hybrids. Stability analysis using both parametric and nonparametric parameters for all hybrids is shown in Table 5 and Table S4. Based on the CV by Francis and Kannenberg, G07, G04, and G02 exhibited high yield performance with low variation. The regression model for stability, using bi and $S^2_{di}$, shows that $b_i = 1$ (not significant) and a low $S^2_{di}$ score indicates a stable hybrid. G02 was the most stable based on bi, followed by G07 and G04, while G01 was the least stable. However, G01 had the smallest $S^2_{di}$, indicating the highest stability by that metric. The stability parameters $\sigma^2_i$ and $W^2_i$ ranked G01, G05, G07, and G04 as the most stable hybrids. EV identified G08 as the most stable, while ASI and ASV ranked G04 and G01 as the most stable. Other AMMI-based statistics (SIPC, MASV, and Za) also ranked G01 as the most stable, with varying rankings for the other hybrids. G01 was the most stable based on the WAASB stability score, followed by G04 and G05.

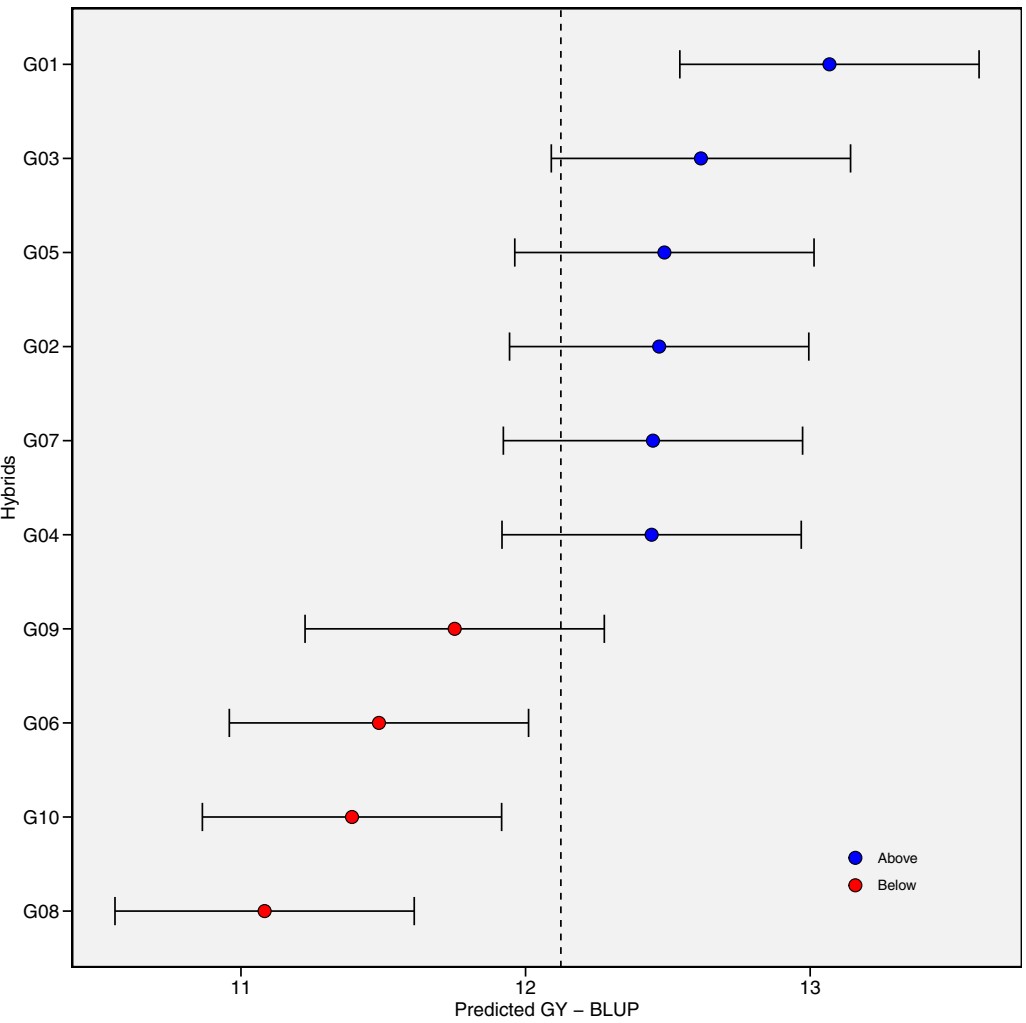

**Figure 3** **The predicted grain yield (GY) performance of the 10 maize hybrids was estimated using best linear unbiased prediction (BLUP).** The vertical dotted line indicates the grand mean, and the horizontal error bars indicate the 95% confidence interval when considering the two-tailed $t$-test.

The Kang stability index ($YS_i$) is a nonparametric index used to identify high yield, stable genotypes. Based on this parameter, G01 was the most stable hybrid, followed by G05 and G03. Similar results were observed in the stability analysis using Huehn's and Nassar and Huehn's statistics ($S^{(1)}$, $S^{(2)}$, $S^{(3)}$, and $S^{(6)}$), with G01 consistently having the smallest values, confirming its stability. However, G03, despite its high GY, was not considered stable according to this parameter. Thennarasu's $NP^{(1)}$ statistics also ranked G01 as the most stable hybrid, but $NP^{(2)}$, $NP^{(3)}$, and $NP^{(4)}$ indicated that G01 was unstable.

The association between mean GY and stability parameters of the maize hybrids was assessed using Spearman's rank correlation (Fig. 4A). GY showed a positive and significant correlation ($p < 0.05$) with $YS_i$ and $S^{(6)}$, but a negative and significant correlation with $NP^{(2)}$ and $NP^{(3)}$. This highlights that the hybrid with the highest GY does not necessarily exhibit stability across environments, according to several stability statistics. The $b_i$ parameter did

Table 5 Rank of the mean grain yield (GY) and stability statistics for the 10 maize hybrids.

| Statistics | Hybrids | | | | | | | | | |
|---|---|---|---|---|---|---|---|---|---|---|
| | G01 | G02 | G03 | G04 | G05 | G06 | G07 | G08 | G09 | G10 |
| GY (t ha$^{-1}$) | 1 | 4 | 2 | 6 | 3 | 8 | 5 | 10 | 7 | 9 |
| CV | 6 | 3 | 10 | 2 | 4 | 7 | 1 | 5 | 8 | 9 |
| $b_i$ | 10 | 1 | 9 | 3 | 5 | 8 | 2 | 4 | 7 | 6 |
| $S^2_{di}$ | 1 | 7 | 10 | 4 | 3 | 5 | 2 | 6 | 8 | 9 |
| $\sigma^2_i$ | 1 | 8 | 10 | 4 | 2 | 6 | 3 | 5 | 7 | 9 |
| $W^2_i$ | 1 | 8 | 10 | 4 | 2 | 6 | 3 | 5 | 7 | 9 |
| EV | 2 | 9 | 8 | 7 | 5 | 6 | 3 | 1 | 4 | 10 |
| ASI | 2 | 8 | 10 | 1 | 3 | 4 | 5 | 7 | 9 | 6 |
| ASV | 2 | 8 | 10 | 1 | 3 | 4 | 5 | 7 | 9 | 6 |
| SIPC | 1 | 10 | 8 | 5 | 4 | 7 | 3 | 2 | 6 | 9 |
| MASV | 1 | 8 | 10 | 3 | 2 | 4 | 6 | 5 | 7 | 9 |
| Za | 1 | 8 | 10 | 2 | 4 | 5 | 6 | 3 | 7 | 9 |
| WAASB | 1 | 7 | 10 | 2 | 3 | 4 | 6 | 5 | 9 | 8 |
| $YS_I$ | 1 | 4 | 3 | 6 | 2 | 8 | 5 | 10 | 7 | 9 |
| $S^{(1)}$ | 1 | 5 | 9 | 2 | 6 | 7 | 4 | 3 | 8 | 10 |
| $S^{(2)}$ | 1 | 4 | 9 | 2 | 5 | 6.5 | 3 | 6.5 | 8 | 10 |
| $S^{(3)}$ | 1 | 6 | 9 | 5 | 4 | 7 | 2 | 8 | 3 | 10 |
| $S^{(6)}$ | 1 | 7 | 5 | 4 | 2 | 9 | 3 | 10 | 6 | 8 |
| $NP^{(1)}$ | 1 | 6 | 9 | 2.5 | 4.5 | 8 | 2.5 | 4.5 | 7 | 10 |
| $NP^{(2)}$ | 9 | 5 | 10 | 7 | 8 | 2 | 6 | 1 | 3 | 4 |
| $NP^{(3)}$ | 10 | 6 | 9 | 5 | 8 | 2 | 7 | 1 | 3 | 4 |
| $NP^{(4)}$ | 10 | 9 | 3 | 5 | 1.5 | 6 | 4 | 1.5 | 7 | 8 |
| ASR | 2.95 | 6.41 | 8.32 | 3.75 | 3.82 | 5.89 | 3.93 | 5.02 | 6.68 | 8.23 |
| RS | 1 | 7 | 10 | 2 | 3 | 6 | 4 | 5 | 8 | 9 |

Note:
Abbreviations: GY, grain yield; CV, Coefficient of variance; $b_i$, Regression coefficient; $S^2_{di}$, Deviation from regression; $\sigma^2_i$, Shukla's stability variance; $W^2_i$, Wricke's ecovalence; EV, Average of the squared eigenvector values; ASI, AMMI stability index; ASV, AMMI stability value; SIPC, Sum of the absolute value of the IPCA scores; MASV, Modified AMMI stability value; WAASB, Weighted average of absolute scores; Za, Absolute value of the relative contribution of IPCAs to the interaction; $YS_i$, Kang's rank sum; $S^{(1, 2, 3, 6)}$, Huehn's and Nassar and Huehn's statistics; $NP^{(1-4)}$, Thennarasu's statistics.

not significantly correlate with any stability statistic except for *CV*. $NP^{(2)}$ and $NP^{(3)}$ displayed a negative correlation with statistical stability, suggesting that hybrid selection based on these parameters differs. The associations between stability statistics are more clearly illustrated in the PCA biplot (Fig. 4B), where three groups of stability parameters were formed. GY, $YS_i$, and $S^{(6)}$ are in Group (G1), while $b_i$, $NP^{(2)}$, $NP^{(3)}$, and $NP^{(4)}$ are in Group 3 (G3), with vectors opposite to G1. Other stability parameters fall in Group 2 (G2), indicating similar hybrid selections. The summary ranking (Table 5) ranks stable hybrids as G01 > G04 > G05 > G07 > G06 > G08 > G02 > G09 > G10 > G03.

## AMMI analysis of grain yield

The AMMI model analysis for GY across 10 hybrids and 10 environments is presented in Table 6, showing that GY was significantly ($p < 0.001$) affected by the environment,

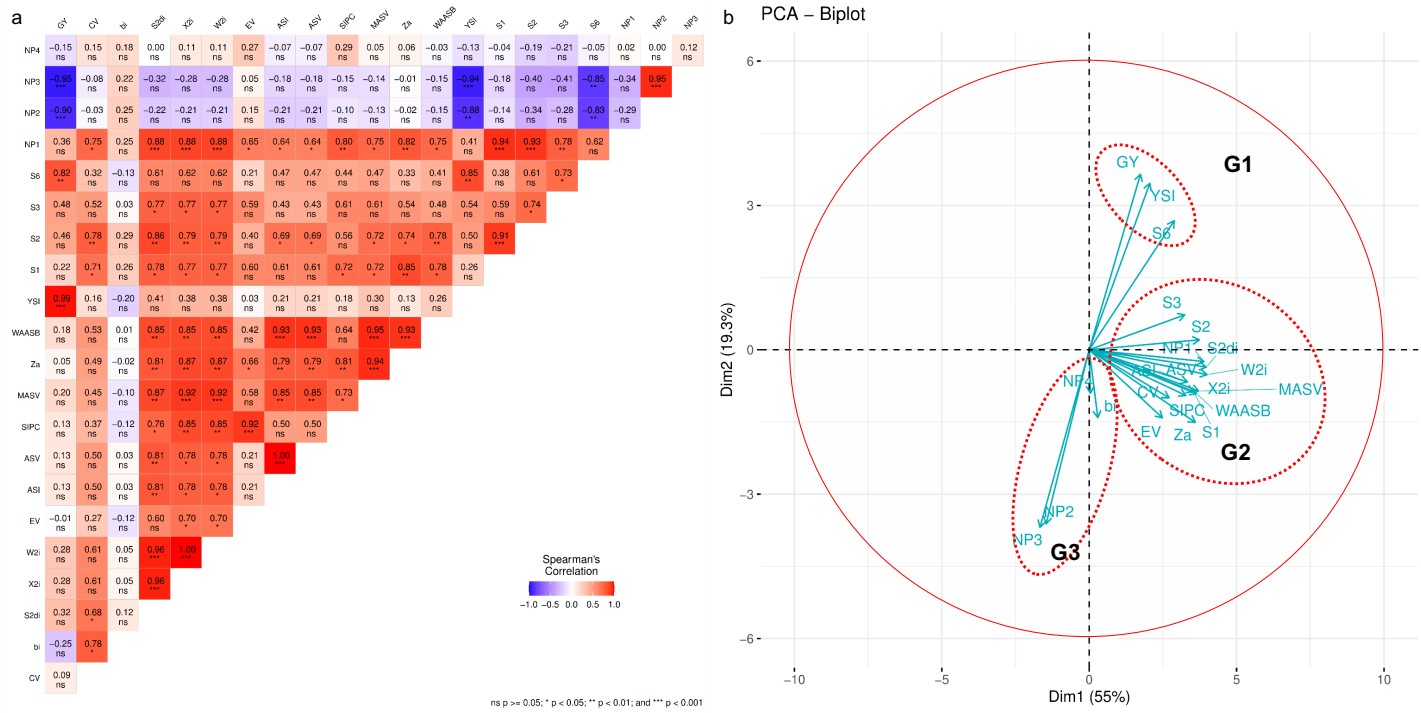

**Figure 4 The association of stability statistics.** (A) Heatmap of Spearman's rank correlation. (B) Principal component analysis (PCA) biplot 20 stability statistics. Asterisks (*, **, and ***) in the heatmap indicate at the significant 0.05, 0.01, and 0.001 probability levels, respectively. See Table 2 for the stability statistics' legends.

genotype, and GEI. This indicates that hybrid responses are influenced by environmental differences, requiring further stability analysis to understand the GEI effect. The GEI was divided into nine interaction principal components (IPCs). Table S5 presents the eigenvalues and proportions of each IPC, with total variance at 5.37. The first 5 IPCs were significant, accounting for 96.20% of the GEI effect, with proportions of 42.20%, 21.90%, 13.50%, 10.80%, and 7.80% from IPC 1 to IPC 5, respectively. The AMMI biplot analysis offers insights into hybrid stability under METs (Fig. 5). In the AMMI1 biplot (Fig. 5A), the hybrid farthest to the right shows the highest GY. Hybrids distant from the abscissa axis suggest stronger environmental impacts on yield, reducing stability. The AMMI2 biplot (Fig. 5B) illustrates the interaction between IPC1 and IPC2, with hybrids near the origin (0.0) showing less influence from GEI and broader adaptability. G01 and G04 being closer to the abscissa axis demonstrated greater stability than other hybrids.

In this study, WAASB statistics were calculated based on mean GY and stability, integrating AMMI analysis and the BLUP model to examine the GEI effect (Fig. 6). The method is displayed graphically, with GY × WAASB divided into four quadrants grouping genotypic means and stability in different environments (Fig. 6B). The predicted WAASBY plot is shown in Fig. 6A. The most unstable hybrids, with a strong GEI influence and high discrimination ability but below-average GY, were in the first quadrant, including G08, G09, G10, E03, and E10. Hybrids in the second quadrant, such as G02, G03, and E07, had high GY but were unstable due to their discrimination powers. The third quadrant contained hybrids with better stability (widely adapted) but lower GY, such as G06 and

**Table 6 Analysis of variance for the additive main effects and multiplicative interaction model (AMMI) of 10 maize hybrids.**

| Source | df | Sum Sq | Mean Sq | Proportion | Accumulated |
|---|---|---|---|---|---|
| E | 9 | 161.15 | 17.91*** | – | – |
| R(E) | 20 | 43.44 | 2.17*** | – | – |
| G | 9 | 143.26 | 15.92*** | – | – |
| G × E | 81 | 145.11 | 1.79*** | – | – |
| PC1 | 17 | 61.21 | 3.60*** | 42.20 | 42.20 |
| PC2 | 15 | 31.76 | 2.12*** | 21.90 | 64.10 |
| PC3 | 13 | 19.61 | 1.51*** | 13.50 | 77.60 |
| PC4 | 11 | 15.70 | 1.43** | 10.80 | 88.40 |
| PC5 | 9 | 11.26 | 1.25* | 7.80 | 96.20 |
| PC6 | 7 | 3.91 | 0.56 ns | 2.70 | 98.90 |
| PC7 | 5 | 1.34 | 0.27 ns | 0.90 | 99.80 |
| PC8 | 3 | 0.30 | 0.10 ns | 0.20 | 100.00 |
| PC9 | 1 | 0.02 | 0.02 ns | 0.00 | 100.00 |
| Residuals | 180 | 89.44 | 0.50 | – | – |
| Total | 380 | 727.51 | 1.91 | – | – |

**Note:**
Abbreviations: E, environment; R, replication; G, genotype; PC, principal component. *, **, and *** in the heatmap indicate at the significant 0.05, 0.01, and 0.001 probability levels, respectively.

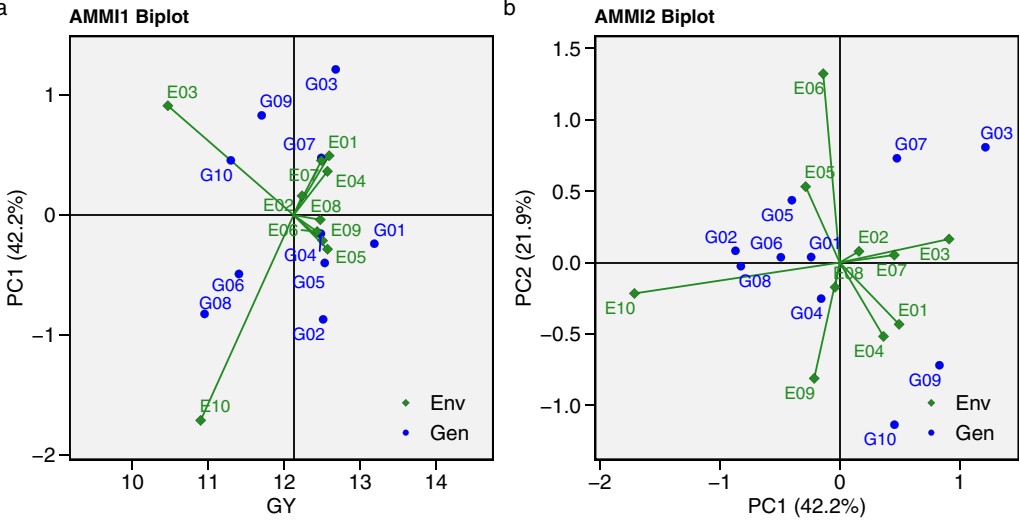

**Figure 5 Additive main effects and multiplicative interaction (AMMI).** (A) The AMMI1. (B) AMMI2 biplots indicate genotype-by-environment interaction for 10 maize hybrids evaluated in 10 environments.

G08. Quadrant four, the best quadrant, included hybrids with high GY and strong stability and adaptability, such as G01, G05, G04, G07, and environments E01, E02, E05, E06, E08, and E09. These environments were highly productive but had low discrimination capabilities.

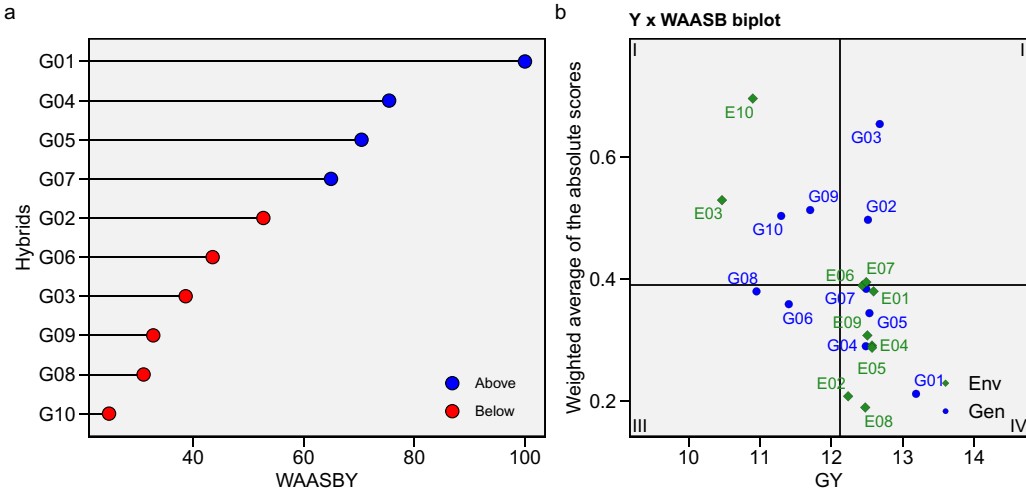

**Figure 6 WAASB statistic.** (A) Estimated values of the weighted average of absolute scores (WAASB) and mean grain yield (WAASBY). (B) Biplot of the mean grain yield and the weighted average of absolute scores for the best linear unbiased predictions of the genotype-*vs.*-environment interaction.

## GGE biplot analysis

GGE biplot analysis was used as an alternative approach to evaluate and identify the ideal, stable, and adaptive GY of maize hybrids across mega-environments. Four types of GGE biplot analyses were applied in this study: discriminativeness *vs.* representativeness, mean *vs.* stability, genotype ranking, and which-won-where (Fig. 7). The length of the environment's vector line from the biplot origin indicates the level of discriminating power in each environment. A longer vector line signifies stronger discrimination in that environment (Fig. 7A). The representativeness of an environment is determined by the angle between the vector line and the average environment coordinate (AEC) axis, with the smallest angle indicating greater representativeness. All environment vector lines exceeded the concentric circles, indicating that all environments were discriminating. E10 had the longest vector lines, meaning hybrids in this environment showed varying GYs. Environments like E03, E05, E06, and E07 had relatively long vector lines, showing strong discriminating power, while the other environments had shorter lines, suggesting similar GYs. According to the AEC axis, E02 and E08 had the smallest angles, making them the most representative environments.

Figures 7B and 7C display the ranking and mean *vs.* stability of the hybrids. Hybrids near the AEC axis had higher mean GYs, while their vertical distances from the AEC axis indicated the magnitude of genotype-by environment effects. G01 had the highest GY, followed by G02, G03, G04, G05, and G07, all of which had above-average grain yields. In the genotype ranking (Fig. 7B), hybrids closer to the center of the concentric circles were ranked higher, with the order being G01 > G07 > G05 > G04 > G02 > G03 > G09 > G06 > G10 > G08. Hybrids with minimal vertical distance from the AEC axis showed less influence from GEI effects. Based on yield performance and stability (Fig. 7C), G01 and G04 had the least deviation from the AEC axis, reflecting high stability across

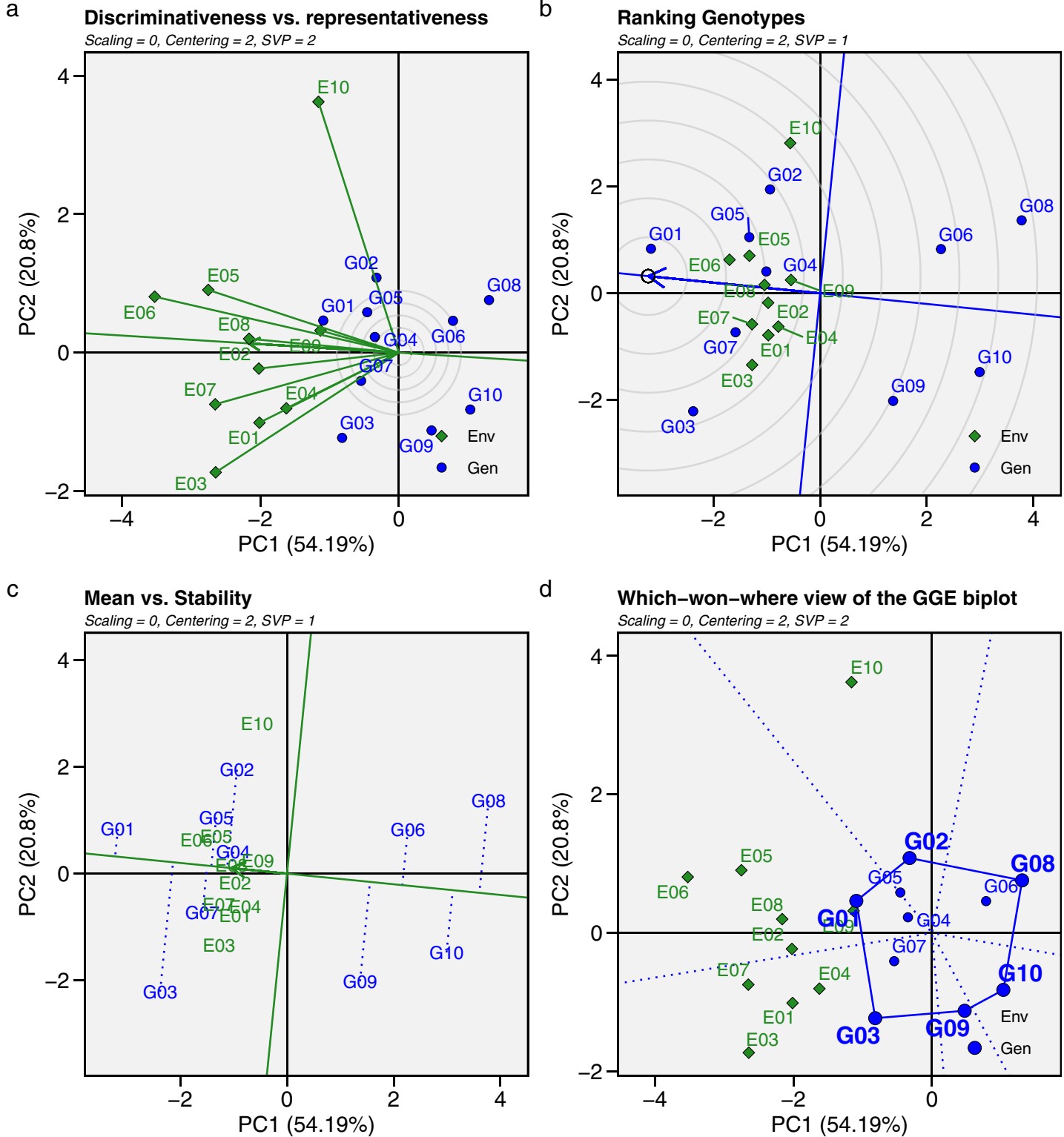

**Figure 7 Genotype + genotype × environment (GGE) biplot.** (A) Discriminativeness *vs.* representativeness. (B) Ranking genotypes. (C) Mean *vs.* stability. (D) Which-won-where view of the GGE biplot for mean grain yield of 10 maize hybrids influenced by genotype × environment interactions across 10 test environments.

environments. G05 and G07 also exhibited excellent stability, demonstrating consistent productivity among most test genotypes.

The which-won-where view of the GGE biplot (Fig. 7D) divides the environments and hybrids into six distinct sectors or regions (marked with dotted lines), each containing different environments, forming groups called mega-environments. Hybrids within each mega-environment show a certain level of adaptation to the environments. Three sectors in the GGE biplot contain environments. The first sector includes E03, E04, and E07; the second sector includes E02, E05, E06, E08, and E09; and the third sector contains only E10. The vertices of the polygon represent the hybrids with the best performance in each mega-environment. G02 performed well and adapted specifically to E10, while G03 and G07 were suited to the first sector, including E01, E03, E04, and E07. G01 and G04 demonstrated broader adaptation across five environments: E02, E05, E06, E08, and E09, while other hybrids did not adapt well to any environment.

## DISCUSSION

### GEI and performance mean of hybrids of maize

In this study, we investigated the GY performance and stability of ten maize hybrids using multiple statistical models across ten different tropical environments. Grain yield, the most important quantitative trait in maize for variety release, is expected to be high and broadly adapted in METs. Genotype (G), environment (E), and GEI significantly influenced grain yield in this study. ANOVA showed that the environment (E) had the largest effect on the total sum of squares, indicating that grain yield is mainly driven by environmental factors. Environmental variance accounted for 35.51% of the total variance, genotype contributed 33.65%, and 30.84% was due to the GEI. The strong environmental effect on maize GY is attributed to abiotic factors arising from climatic and edaphic variations across growing locations (Table 1). Various environmental factors such as agroecosystems, agroclimate, altitude, and soil conditions can cause genotype-environment interactions, leading to variations in hybrid response (yield and grain quality) in METs (*Mafouasson et al., 2018*).

In terms, the heritability of GY reached 0.34 (34%), indicating that environmental factors have a large influence on GY. However, genotypic heritability (mean basis) tended to be high, reaching 0.89 (89%). Heritability of entry mean basis is defined as the ratio of the genetic variance component to the phenotypic variance on a hybrid mean basis using random effect using the best linear unbiased prediction (BLUP) method. The higher entry mean basis heritability indicates a strong predictability of genotype performance across diverse environments (*Gela et al., 2023*). However, many factors, including the significance of GEI, can influence the magnitude of heritability, suggesting that these values should be interpreted within the context of experimental conditions.

G01 (R0211) had the highest mean GY performance in METs, reaching 13.19 t ha$^{-1}$, followed by G03 (12.68 t ha$^{-1}$), G05 (12.53 t ha$^{-1}$), G02 (12.51 t ha$^{-1}$), and G07 (12.49 t ha$^{-1}$). BLUP predictions confirmed G01 as having the highest mean GY, while G08 had the lowest. The BLUP method is considered accurate for estimating GY, as it optimizes predictions from random effects when mixed linear effects are present (*Olivoto et al., 2019*). Similar results have been reported in studies predicting genotype performance and

variance components in maize and other crops (*Baretta et al., 2016*; *Gela et al., 2023*; *Shobeiri, Pezeshkpour & Naseri, 2024*). G01, G03, G05, G02, G07, and G04 had mean GYs above the overall average, with genotype accuracy (*As*) reaching 94.2%. E01 was the environment with the highest mean grain yield across all hybrids, while E10 had the highest CV. Most maize hybrid evaluations, including this study, focus on GEI interactions. The importance of GEI in influencing GY highlights the need for complementary stability analysis to clarify interactions, assess yield potential, and determine stable hybrids adaptable to specific environments (*Mafouasson et al., 2018*; *Singamsetti et al., 2021*; *Engida et al., 2024*).

## Assessment of the grain yield stability under METs

In this study, various stability statistics and models, including parametric and nonparametric analyses, were used to assess the GY stability of hybrids under METs. Using multiple stability statistics simultaneously is considered more accurate and effective for identifying stable hybrids than relying on a single stability measure. Other researchers have also used different stability statistics to select stable, high-yielding crops, including maize (*Ruswandi et al., 2022*; *Wicaksana et al., 2022*), chickpea (*Karimizadeh et al., 2023*), oats (*Kebede et al., 2023*), and other crops. Each stability statistic has its own criteria and model for determining a stable genotype, with strengths and weaknesses in each parameter (*Gela et al., 2023*). The stability statistics used in this study represent the most important methodologies for evaluating yield stability across environments, such as the regression coefficient, Shukla's stability, the AMMI stability index and value (ASI and ASV), Kang's rank sum, and newer methods like the weighted average of absolute scores (WAASB). Evaluating the 10 hybrids in Indonesia's tropical region using this comprehensive approach provides breeders with advanced insights to make more informed decisions when selecting hybrids for variety release based on their stability patterns.

Spearman's rank correlation coefficient was calculated for each pairwise comparison of the stability parameters and mean GY. The study found that GY was strongly and positively correlated with $YS_i$ and $S^{(6)}$, suggesting that high-yielding hybrids tend to be stable according to these two stability statistics. However, GY showed no significant correlation with most stability statistics, indicating that hybrids with high yields may not necessarily perform stably under METs. G03, with a high yield of 12.68 t ha$^{-1}$, was found to be sensitive to environmental changes, as indicated by its diverse yield values across environments. In contrast, G07 and G04, ranked fifth and sixth in mean yield performance, were identified as stable hybrids based on the summary ranking of all stability statistics. G01, which had the highest GY, was consistently recognized as the most stable hybrid across all stability statistics. According to the summary ranking, G01, G04, G05, and G07 were the stable hybrids.

Most of the stability statistics calculated in this study showed strong positive correlations with each other, except for $b_i$, $NP^{(2)}$, $NP^{(3)}$, and $NP^{(4)}$, which predominantly had negative correlations with GY and other stability statistics. The PCA biplot depicted these parameters separately, suggesting that hybrids deemed stable based on these measures were not typically high-yielding. The stability statistics were divided into three

groups based on the PCA biplot and their characteristics in genotype ranking. Group I (G1) included GY, $YS_i$, and $S^{(6)}$, which, based on Spearman correlation, were significantly and positively correlated, indicating that breeders or agronomists could simultaneously select hybrids using $YS_i$, and $S^{(6)}$ for developing superior, consistently performing genotypes. Group II (G2) contained the remaining stability statistics, while $b_i$, $NP^{(2)}$, $NP^{(3)}$, and $NP^{(4)}$ were grouped in G3. Stability methods are often classified into static and dynamic concepts based on their relationship to GY performance. The static (or biological) concept posits that a stable genotype maintains consistent yields despite environmental variation, with yield performance close to zero across diverse environments. In contrast, the dynamic (or agronomic) concept suggests that genotypes respond predictably to environmental changes, following the same trend as the average genotype response, thus accounting for GEI (*Leon, 1985*; *Becker & Leon, 1988*). Static stability is more beneficial for resource-limited farmers, who prioritize genotypes that yield consistently under minimal inputs. Breeders and agronomists, however, prefer dynamic stability to identify genotypes with high yield potential and strong responses to optimal management (*Kebede et al., 2023*). Ultimately, the classification of stability, whether static or dynamic, depends on the data and testing environments, which influence its relationship to yield performance (*Pour-Aboughadareh et al., 2022*). Based on Spearman's rank correlation and the PCA biplot, G1 is classified as dynamic stability, while G2 and G3 represent static stability.

## AMMI and graphical biplot analysis for the identification of stable and adaptive genotypes

The AMMI model and GGE biplot are highly effective statistical tools widely used to analyze GEI effects. The key difference between them is that GGE includes both the genotype's main effect and the G × E interaction, whereas AMMI focuses solely on the G × E interaction (*Li et al., 2023*; *Dang et al., 2024*; *Kona et al., 2024*). In this study, the AMMI model partitioned GEI effects into nine IPCs, with the first five IPCs being significant, explaining a total of 96.20% of the GEI effect. These significant IPCs serve as a foundation for further stability analysis in AMMI modelling (*Gauch, 2013*). *Gauch & Zobel (1996)* demonstrated that projecting the AMMI model with the first two significant IPCs is accurate, and in this study, the detection of two significant IPCs was sufficient to identify the superior genotype. The total variance explained by IPC1 and IPC2 reached 64.1% in the AMMI biplot model.

The AMMI model was developed to assess significant multiplicative interactions between genotypes and environments, focusing on the AMMI1 and AMMI2 biplots. The AMMI1 biplot is used to evaluate hybrids' potential GY and stability under METs. In the AMMI1 biplot the ordinate represents environmental effects, while the abscissa shows the mean GY. Genotypes positioned farther from the abscissa axis are more influenced by environmental factors (*Azrai et al., 2023*; *Wang et al., 2023*). Based on AMMI1, G01, located farthest to the right, had the highest GY, while G04 experienced less environmental influence. The AMMI2 biplot explains the GEI through IPC1 and IPC2. The distance from the origin indicates the extent of GEI effects on the genotype, with greater distance reflecting stronger GEI influence (*Habtegebriel, 2022*). G01 and G04 were identified as

widely adapted, environmentally stable hybrids based on their GY performance, while G03, G09, and G10 had weaker stability due to their greater distance from the AMMI2 biplot origin.

Model diagnosis is crucial in selecting the best model for a dataset (*Gauch, 2013*). In this study, the AMMI model was integrated with the BLUP model, which demonstrated the best performance in predicting genotype response. By incorporating all IPCs from AMMI into the BLUP method, we accurately measured genotype stability, leveraging the strengths of both models using WAASB. This approach simultaneously assesses performance and stability, providing valuable insights into genotype distribution and environmental effects (*Olivoto et al., 2019*). Similar findings have been reported in studies on maize (*Huang et al., 2021*), soybean (*Nataraj et al., 2021*), and faba bean (*Gela et al., 2023*). Based on WAASB, G01, G04, G05, and G07 were identified as stable, high-yielding hybrids, aligning with the ranking summary of all stability statistics.

The GGE biplot is a powerful and versatile tool for evaluating GEI and recommending genotypes for specific environments (*Yan & Kang, 2003*). This model combines genotype main effects and GEI effects, with eigenvalue decomposition performed simultaneously for both, emphasizing the principal component that explains the most variance (*Bocci et al., 2020*; *Dang et al., 2024*). This study analyzed four types of GGE: discriminativeness *vs.* representativeness, mean *vs.* stability, ranking genotypes, and which-won-where. The GGE biplot identifies the environment with the greatest discrimination between genotypes (marked by the longest vector line) and the most representative environment (marked by the smallest angle with the AEC axis) (*Esan et al., 2023*). E10 had the longest vector, indicating strong discriminating power, while E02 and E08 had the lowest angles, making them the most representative environments. The variation in genotype responses at E10 (Talakar) could be due to its distinct agroecological and climatic conditions, with annual rainfall reaching 2,030 mm and relatively low temperatures. Such environmental differences pose challenges for stability studies. In contrast, E02 (Bantul) and E08 (Jombang) have more representative conditions, with rainfall between 1,200 and 1,400 mm, which are ideal for growth and achieving optimal results (*Gyamerah et al., 2023*). Thus, E02, E08, and similar environments are suitable for selecting superior genotypes, while E10 had a stronger ability to differentiate phenotypic expression in maize hybrids in this study.

Genotype stability based on the GGE biplot was determined by the vertical distances of the hybrids from the AEC axis. Hybrids with the smallest distance to the AEC axis were considered stable across METs (*Wang et al., 2023*). Similar to the AMMI model, the GGE biplot identified G01 and G04 as the most stable hybrids across environments, followed by G05 and G07. A key advantage of the GGE biplot is its ability to identify genotypes that are specifically adaptive to particular environments using the "which-won-where" perspective. This biplot divides environments into mega-environments, grouping genotypes that are adaptive to a specific environment in the same sector (*Hossain et al., 2023*). In this study, G01, G04, and G05 were widely adapted to five environments: E02, E05, E06, E08, and E09. G03 and G07 were suited to E01, E03, E04, and E07, while G02

adapted well to E10. However, the remaining hybrids (G06, G08, G09, and G10) had no specific environment in which they performed well, indicating poor performance in some or all test environments. Maize hybrids with high yield performance or those adapted to specific environments can become superior in this region, which could significantly boost the income of maize farmers by maximizing genotype potential in the appropriate environment.

## Comparison of the stable hybrids based on all the stability statistics and graphical biplot

This study demonstrates that some hybrids exhibit stable GY performance according to certain stability parameters, while appearing unstable according to others. This variability is a common challenge in GEI and yield stability studies. Each statistical stability parameter uses distinct criteria for selecting stable hybrids, offering valuable insights for genotype selection. Choosing the best genotype based solely on the highest mean GY may not provide sufficient information for selecting a stable hybrid. For instance, G03 (R0654) ranked second in mean GY (12.68 t ha$^{-1}$), but several stability parameters classified it as unstable due to its high variability across test environments. Grouping stability parameters helps breeders and agronomists select hybrids tailored to the specific requirements of each stability method. Stability parameters correlated with GY support the selection of both superior and consistently performing genotypes. Additionally, the AMMI and GGE biplots offer advanced insights into the stability and adaptability of each hybrid across test environments, providing a broader perspective for selecting hybrids suited to specific environments.

The differences in stable hybrids across various stability parameters can be addressed using the average summary ranking (ASR), where a lower ASR value indicates greater stability. Our findings identified four hybrids—G01 (R0211), G04 (R0105), G05 (R0118), and G07 (R0641) as the most stable based on the ASR, as well as through multivariate analysis (AMMI and GGE biplot) and the superiority index WAASBY. However, this study focused solely on GY performance. Exploring multiple traits that align with the preferences of farmers, seed producers, and consumers is a valuable consideration for future breeding strategies (*Dermail et al., 2022*). High GY performance and stability, aligned with breeding goals and cultivar recommendations, provide these hybrids with the potential to become superior varieties of tropical maize in Indonesia. G01, G04, G05, and G07 are new superior hybrid candidates released by PT. RAJA and are currently being proposed as new varieties at the Center for Plant Varieties Protection and Agricultural License (PVTPP), Ministry of Agriculture, Indonesia. The development of these hybrids began in 2018, using cross populations of commercial hybrid varieties commonly grown in Indonesia (BISI 18, NK212, DK95, ADV777, and others), and they were carefully selected to produce new hybrids. In addition to their high and stable yields, these four hybrids also demonstrated strong agronomic performance, disease resistance, and high-quality yield content. Overall, they could be promising genetic resources for improving and stabilizing maize hybrid grain yields in tropical regions.

## CONCLUSIONS

The environment, genotype, and GEI effects significantly contributed to the overall variance in the GY performance of maize hybrids in the METs. Stability analysis, combining stability statistics and multivariate methods (AMMI and GGE biplots), proved more accurate in predicting and identifying stable and adaptive hybrids. The BLUP prediction identified G01 as having the highest GY. Four hybrids—G01 (R0211), G04 (R0105), G05 (R0118), and G07 (R0641)—were the most stable across all stability parameters used in this study. Therefore, they can be recommended as new superior varieties and valuable genetic resources for maize development programs in tropical regions of Indonesia.

## ACKNOWLEDGEMENTS

The authors thank the research and development team of PT Restu Agropro Jayamas (RAJA), Indonesia for providing plant material and for technicians at the experimental stations for helping with the experiments.

### Funding

This study was supported by PT Restu Agropro Jayamas-RAJA, Indonesia. The research and development team of PT Restu Agropro Jayamas (RAJA), Indonesia provided plant material and the technicians at the experimental stations helped with the experiments. The funders had no role in study design, data collection and analysis, decision to publish, or preparation of the manuscript.

### Grant Disclosures

The following grant information was disclosed by the authors:
PT Restu Agropro Jayamas-RAJA, Indonesia.

### Competing Interests

The authors declare that they have no competing interests.

### Author Contributions

- Dedy Supriadi conceived and designed the experiments, performed the experiments, authored or reviewed drafts of the article, and approved the final draft.
- Yusuf Mufti Bimantara conceived and designed the experiments, performed the experiments, analyzed the data, prepared figures and/or tables, authored or reviewed drafts of the article, and approved the final draft.
- Yuniel Melvanolo Zendrato conceived and designed the experiments, analyzed the data, prepared figures and/or tables, authored or reviewed drafts of the article, and approved the final draft.
- Eko Widaryanto conceived and designed the experiments, authored or reviewed drafts of the article, and approved the final draft.

- Kuswanto Kuswanto conceived and designed the experiments, authored or reviewed drafts of the article, and approved the final draft.
- Budi Waluyo conceived and designed the experiments, authored or reviewed drafts of the article, and approved the final draft.

## Data Availability

The raw measurements are available in the Supplemental File.

## Supplemental Information

Supplemental information for this article can be found online at http://dx.doi.org/10.7717/peerj.18624#supplemental-information.

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
