# Peer review of "Assessment of genotype by environment and yield performance of tropical maize hybrids using stability statistics and graphical biplots"

_PeerJ, doi:10.7717/peerj.18624_

## Round 0.1 · original submission · Major Revisions

· Academic Editor

Major Revisions

Reviewers have made the necessary points. Genotype selection is important. However, discuss the effectiveness of the models in generalizing your results.

Reviewer 1 ·

Basic reporting

The study focuses on evaluating genotype-by-environment interactions (GEI) and the performance stability of maize hybrids across multiple environments, which is a relevant and critical issue in plant breeding. The topic is well-chosen, addressing the real-world challenges in improving maize varieties for tropical environments. The use of statistical models such as AMMI and GGE biplots, combined with stability statistics, is commendable for its robust methodology and comprehensive data analysis. This manuscript demonstrates strong scientific merit, but refinement in writing, structure, and the presentation of statistical findings could significantly enhance its impact. It is recommended to Improve clarity by reducing redundancy, clarifying ambiguous sentences, and providing more direct explanations of statistical methods. Refine grammar with attention to articles, verb tense consistency, and sentence structure. Streamline references, ensuring they are directly relevant and not repeated unnecessarily. Enhance practical implications by linking statistical findings more directly to real-world breeding practices and applications.
In abstract, The term "YSi and S(6)" appears without explanation. Readers unfamiliar with these terms might find it confusing. Defining these metrics briefly within the abstract or using more descriptive terms would enhance readability. A more concise description of the importance of GEI and stability analyses would help streamline the abstract. Moreover, add key values of the results obtained. Please add the potential impact or application of these findings on maize breeding and adaptation to tropical environments in the conclusion would improve the abstract's appeal. Ensure all terminology used in the abstract (e.g., GEI, AMMI, BLUP) aligns perfectly with terms and abbreviations introduced in the main body.
In introduction, some of the references (e.g., Ranum et al., 2014) are outdated. Citing more recent studies, especially those related to stability statistics or tropical maize improvement, would strengthen the literature foundation. Some references, like those to FAOSTAT (2022) and Ranum et al. (2014), provide valuable information, but the link between these facts and the research focus must be strengthened. The introduction jumps between topics (e.g., maize importance, GEI challenges) without clear transitions. Please avoid repetition of points. While the importance of GEI is mentioned, the specific gap your study addresses could be highlighted more clearly. The objective at the end of the introduction could be more specific. Clarifying that the study intends to compare different stability models and their combined effectiveness in evaluating maize hybrids would give a stronger direction to the readers.
In methods few areas could be revised for precision and clarity. The description of the commercial varieties as "checks" could be explained further. Are these checks well-established in similar environments? Regarding statistical analysis, it would also help to provide a brief explanation of why certain stability parameters were selected or favored in this study. Describing the appropriate rationale for combining different models (AMMI, GGE biplot, etc.) early in the introduction section would help readers understand the approach better.
There is no explicit discussion of why the random effects model was selected. The decision to treat the effects as random should be justified, possibly with reference to the objectives of broader inference. The authors should discuss that how the different heritability estimates (34% for plot-level vs. 89% for mean basis) affect the interpretation of the results. For example, higher heritability on a mean basis suggests strong predictability of genotype performance across environments, but its relationship with GEI should be discussed more in-depth. The coefficient of variation (CV) is mentioned to reflect the accuracy of the data. While a CV of 5.81% is considered low and indicative of data reliability, the CV can sometimes be misleading, especially in a multi-environment trial where variations could come from specific environments or genotypes. Thus, reporting the CV alone may not fully capture the variability introduced by GEI, and further discussion on environmental and genotype-specific variance should be considered.
The study uses an extensive set of stability parameters (15 in total), including both parametric (e.g., bi, S²di) and non-parametric (e.g., Kang’s stability index, AMMI-based parameters) methods. These large number of parameters could make interpretation overwhelming. Do all these parameters provide unique insights, or are some redundant? A clearer explanation of the relative importance of each stability measure, or a strategy for selecting the most informative ones, would strengthen the analysis. Certain sections are overly long, and key points become buried in excessive detail, particularly in the results section.
Please justify that why you used Spearman’s Rank Correlation to evaluate the association among stability parameters rather than other multivariate approaches? The interpretation of the IPCs should be made more explicit. For instance, explaining what each significant IPC represents in terms of environmental or genotypic effects would aid in practical interpretation. The use of the GGE biplot to visualize the "which-won-where" pattern and genotype ranking across environments is briefly mentioned. However, the discriminativeness and representativeness of environments should be more clearly discussed. Specifically, the study should elaborate on how these traits guide the selection of environments for future trials or breeding programs. For example, which environments represent the largest challenge for stability and why? While the accuracy of genotype selection (As = 0.94) is high, there is little discussion on the limitations of BLUP, especially concerning the assumptions of the mixed model. How sensitive are the BLUP estimates to the assumptions of homogeneity of variance, and how do these estimates perform in the presence of high GEI? It would be useful to clarify why both the “metan” package and PBSTAT-GE were used for stability statistics estimation. Was there a specific reason to use both platforms, and did the results from the two align? While the figures provide useful information, the text should guide the reader more explicitly in interpreting these plots. The violin plots could benefit from a more detailed explanation of the observed variation within and across environments. Please improve the statistical analysis in terms of method justification, interpretation of complex stability parameters, and the integration of results from different analytical approaches. Additionally, some redundancy in stability analysis could be reduced, and the discussion on environmental representativeness and implications for breeding should be expanded.
Specific Comments:
112-114: clarify the sentence for better readability, “Multilocation testing was out simultaneously from March to October 2023 at ten center locations 113 of maize in Indonesia, namely in Klaten (E01), Bantul (E02), Tuban (E03), Boyolali (E04), 114 Nganjuk (E05), Blitar (E06), Kediri (E07), Jombang (E08), Malang (E09) and Takalar (E10).”
115-116: The sentence is not complete and it is not clear what does authors intends to convey here: “which affect the growth and development of crops in the study areas.”
Line Nos. 116-120: The description provided in these lines lacks sufficient detail to fully understand the experimental setup. The current explanation mentions "C2 to D3 for climate types" but does not clearly define which alphabet corresponds to which climate type. It is essential to explicitly link each alphabet (C2, D3, etc.) to a specific climate classification and provide the main environmental characteristics, such as temperature ranges, humidity levels, wind speed, and average rainfall. Additionally, more comprehensive details regarding the soil types (grumosol, ultisol, alluvial, andosol, and inceptisol) are required. Key characteristics of these soils, including their texture, organic matter content, water retention capacity, and nutrient availability, should be included. This information is critical for evaluating the reliability of the data and the accuracy of the results, as soil and climate conditions significantly affect experimental outcomes. Without this, it is difficult to predict how reproducible or valid the findings are under different environmental conditions.
Mention the details of management practices that were adopted in Line No. 131-133: “All appropriate crop management practices were implemented according to the recommended guidelines at each site.”
133-134: Expand this information for clarity “Weeding was performed by cleaning the weeds around the plants and mulching was done by elevating the mounds and loosening the soil to create better soil aeration.”
In line 343, it would be more grammatically correct to say, “The development of superior maize hybrids is the goal of breeding programs...” instead of “Development of superior maize...” This kind of article use is frequent throughout the paper and should be reviewed.
The sentence in lines 349–350, “ANOVA of the current study revealed that the significance of E has a large proportion of the G and GEI effects...” is unclear. Clarify it.
In line 364, it would be clearer to say, “G01 had the highest mean grain yield...” rather than “G01 had the best mean grain yield performance...” The phrase "best mean grain yield" is awkward and could be better phrased as "highest mean grain yield."
The section on "Assessment of grain yield stability under METs" (lines 376–383) is valuable but could be more engaging if it included more practical implications for breeders and farmers. While the statistical detail is thorough, connecting this analysis more directly to its impact on breeding programs would strengthen the relevance.
The PCA biplot discussion (lines 403–419) and the classification into groups (G1, G2, G3) are strong but could benefit from a more straightforward summary of the practical importance of each group's classification. The detailed statistics are well-presented, but the main takeaway regarding how this information helps in selecting superior hybrids could be more concise.
In line 421, the comparison of AMMI and GGE biplots could be streamlined into a smoother sentence without excessive use of parentheses, making the explanation clearer.
The final sentence of the conclusion (lines 518–520) feels overly general and could be more impactful by highlighting a specific recommendation or future direction.
The caption for Figure 1, "Plot of maize hybrids vs. ten environmental conditions," lacks clarity and does not adequately describe the data presented. All figure captions should be self-explanatory, providing sufficient detail so that the figure can be understood without reference to the main text. Specifically, ensure that the figure caption fully explains what the figure is intended to convey.
Furthermore, it is important to include the complete form of all abbreviations in the footnotes of each figure for clarity.
In Table 9, the principal component analysis (PCA) results are presented; however, the eigenvalues of all nine principal components (PCs) are not provided. Including the eigenvalues is essential for understanding the contribution and importance of each principal component in explaining the variance within the dataset. The eigenvalues will help assess how much of the total variance is captured by each PC and provide a clearer picture of the dimensionality reduction. Please include the eigenvalues for all nine PCs to offer a more comprehensive interpretation of the PCA results.

Experimental design

While the RCBD is an appropriate choice for the experiment, several areas need further clarification or justification to enhance the reliability of the experimental design. These include the handling of environmental variability, site-specific management practices, specify whether uniformity in physiological maturity was achieved across the sites before harvesting and if adjustments were made to account for any discrepancies, provide more detailed information on how environmental factors such as soil moisture, temperature, and rainfall were monitored and controlled. The plot size (14 m²) is reasonable, but no mention is made of buffer zones or border rows to minimize edge effects.

Validity of the findings

Findings can be valid provided the authors clearly provide the queries in experimental design.

Reviewer 2 ·

Basic reporting

A lot of effort has gone into this and it is an original study, but there is no supplementary table associated with yield or field parameters. It should be related to yield and parameters in the field trial in the Results section. Also, there is almost no discussion section. Very weak. It should be enriched . Other comments are stated on the main manuscript in PDF format.

Experimental design

-

Validity of the findings

-

Additional comments

-

Annotated reviews are not available for download in order to protect the identity of reviewers who chose to remain anonymous.

Reviewer 3 ·

Basic reporting

Your manuscript is a valuable scientific study in terms of comparing GEI parameters, which have an important place in plant breeding studies. However, you need to revise your manuscript by paying attention to the following points.
1. The title should be corrected.
2. The English of the manuscript should be corrected by a professional native speaker.
3. Line 33: It is not appropriate to start a sentence directly with an abbreviation. Please write the long version first.
4. The introduction section is well organized but should be improved in terms of English language.
5. Line 116: What do you mean by C2 and D3 binary types? You should write the Material Method section more explanatory.
6. Line 119: What is (masl)?
7. Line 120: Table 1 is not explanatory enough. In order to talk about the stability of genotypes, we need to make sure that you are fertilizing according to the soil analysis results. Because agronomic practices will also affect their growth, development and yield.
8. More information is needed about your agronomic practices. Are you sure that fertilization and irrigation management will be the same everywhere? Climate data is also missing. Total precipitation and average temperature values ​​should be given.
9. Line 151 Fifteen note 15
10. Line 210: What is the unit?
11. The first paragraph of the discussion section is a repetition of the introduction and your findings, and the references are not placed in the appropriate places. I think you need to reorganize the discussion section.
12. The conclusion section is like a repetition of the summary.
13. I read too many repetitive sentences throughout the text. This makes the writing long and boring. Please simplify it. In the conclusion section, you should emphasize the effectiveness of the GEI parameters and which ones can be used. Presenting the selection of superior genotypes in your region will not have a global effect. The point you need to emphasize for this is different. Besides, it would be a more accurate approach to discuss the effectiveness of the parameters throughout the manuscript.

Experimental design

No comment

Validity of the findings

No comment

Additional comments

No comment

Annotated reviews are not available for download in order to protect the identity of reviewers who chose to remain anonymous.

---

## Round 0.2 · accepted · Accept

· Academic Editor

Accept

Your corrections are acceptable for your manuscript to be accepted.

Reviewer 1 ·

Basic reporting

I am pleased to see that the authors have successfully addressed all the revisions and recommendations previously suggested for the manuscript titled "Assessment of genotype by environment and yield performance of tropical maize hybrids using stability statistics and graphical biplots." The manuscript has undergone substantial improvements, particularly in the clarity of the results and the discussion of key findings.

Experimental design

The authors have successfully addressed the ambiguities in experimental design and the revisions contribute significantly to a better understanding of the experimental design and methodology.

Validity of the findings

The findings appear robust and credible, with the data supporting the study's conclusions effectively.

Reviewer 2 ·

Basic reporting

The corresponding author has implemented our revision suggestions. There is no objection on my part for its publication

Experimental design

-

Validity of the findings

-

Additional comments

-

Reviewer 3 ·

Basic reporting

The corrections I specified have been made. Thank you

Experimental design

The corrections I specified have been made. thank you

Validity of the findings

The corrections I specified have been made. thank you

Additional comments

The corrections I specified have been made. thank you